# The Global Jukebox: A public database of performing arts and culture

Anna L. C. Wood[1,2]*, Kathryn R. Kirby[3,4], Carol R. Ember[5], Stella Silbert[1], Sam Passmore[6,7], Hideo Daikoku[8], John McBride[9], Forrestine Paulay[1,10], Michael J. Flory[11], John Szinger[1], Gideon D'Arcangelo[12], Karen Kohn Bradley[10], Marco Guarino[13], Maisa Atayeva[14], Jesse Rifkin[1], Violet Baron[15], Miriam El Hajli[1], Martin Szinger[1], Patrick E. Savage[6]*

1 Association for Cultural Equity (ACE), Hunter College, New York City, NY, United States of America, 2 Centro Studi Alan Lomax, Palermo, Italy, 3 Department of Linguistic and Cultural Evolution, Max Planck Institute for the Science of Human History, Jena, Germany, 4 Department of Ecology and Evolutionary Biology, University of Toronto, Toronto, Canada, 5 Human Relations Area Files at Yale University, New Haven, CT, United States of America, 6 Faculty of Environment and Information Studies, Keio University, Fujisawa, Japan, 7 Evolution of Cultural Diversity Initiative, School of Culture, History and Language, College of Asia and the Pacific, The Australian National University, Canberra, Australian Capital Territory, Australia, 8 Graduate School of Media and Governance, Keio University, Fujisawa, Japan, 9 Center for Soft & Living Matter, Institute for Basic Science, Daejeon, South Korea, 10 Laban/Bartenieff Institute for Movement Studies, New York, NY, United States of America, 11 Research Design and Analysis Service, New York State Institute for Basic Research in Developmental Disabilities, Staten Island, NY, United States of America, 12 Arup, New York, NY, United States of America, 13 American Studies Program, University of Texas, Austin, Texas, United States of America, 14 Paul H. Nitze School of Advanced International Studies, Johns Hopkins University, Baltimore, MD, United States of America, 15 University of Indiana, Folklore & Ethnomusicology, Bloomington, Indiana, United States of America

* annalwood@gmail.com (ALCW); psavage@sfc.keio.ac.jp (PES)

**Data Availability Statement:** All coded data are available at https://github.com/theglobaljukebox. Source code for data conversion and analysis are available at https://zenodo.org/record/6537663#.YnszmllS_BK. Audio files are available for

## Abstract

Standardized cross-cultural databases of the arts are critical to a balanced scientific understanding of the performing arts, and their role in other domains of human society. This paper introduces the Global Jukebox as a resource for comparative and cross-cultural study of the performing arts and culture. The Global Jukebox adds an extensive and detailed global database of the performing arts that enlarges our understanding of human cultural diversity. Initially prototyped by Alan Lomax in the 1980s, its core is the Cantometrics dataset, encompassing standardized codings on 37 aspects of musical style for 5,776 traditional songs from 1,026 societies. The Cantometrics dataset has been cleaned and checked for reliability and accuracy, and includes a full coding guide with audio training examples (https://theglobaljukebox.org/?songsofearth). Also being released are seven additional datasets coding and describing instrumentation, conversation, popular music, vowel and consonant placement, breath management, social factors, and societies. For the first time, all digitized Global Jukebox data are being made available in open-access, downloadable format (https://github.com/theglobaljukebox), linked with streaming audio recordings (theglobaljukebox.org) to the maximum extent allowed while respecting copyright and the wishes of culture-bearers. The data are cross-indexed with the Database of Peoples, Languages, and Cultures (D-PLACE) to allow researchers to test hypotheses about worldwide coevolution of aesthetic patterns and traditions. As an example, we analyze the global relationship between song style and societal complexity, showing that they are robustly related, in

streaming at http://theglobaljukebox.org, with some restrictions as explained in the text. The datasets are archived with ZENODO, and the DOI provided by ZENODO should be used when citing particular releases of Global Jukebox datasets, which are available within the respective GitHub repositories. For details regarding third-party streaming and downloading of audio, see Section 2.6 ("Availability of Audio Recordings").

**Funding:** The Global Jukebox has been developed with support from the National Endowment for the Arts, the National Endowment for the Humanities, the Concordia Foundation, the Rock Foundation, and Odyssey Productions. PES, HD, and SP are supported by funding from the Yamaha corporation, a Grant-in-Aid from the Japan Society for the Promotion of Science (#19KK0064), and by grants from Keio University (Keio Global Research Institute and Keio Gijuku Academic Development Fund). The funders had no role in study design, data collection and analysis, decision to publish, or preparation of the manuscript.

**Competing interests:** The authors have declared that no competing interests exist.

contrast to previous critiques claiming that these proposed relationships were an artifact of autocorrelation (though causal mechanisms remain unresolved).

## Introduction

During the 20th century, anthropologists began organizing data on cross-cultural diversity in ways that could be systematically compared on a global scale. The *Ethnographic Atlas* [1] coded data on social structure, kinship, religion, and economy; the Human Relations Area Files (HRAF) [2] compiled and subject-indexed detailed ethnographic texts; Ethnologue [3] and Glottolog [4] cataloged linguistic diversity. These resources allow scientists to quantitatively test cross-cultural hypotheses using global data. The 21st century has seen a resurgence of interest in such global databases, stimulating new research and debate on the nature of cross-cultural diversity and cultural evolution [5–12]. Alan Lomax and Conrad Arensberg's Expressive Style Research Project at Columbia University, which also began in the mid-20th century, complemented resources like the *Ethnographic Atlas* for the domain of the performing arts by integrating its cross-cultural classification design into the research design of "Cantometrics" ("canto" = song, "metrics" = measure). In the 1980s their studies of song, dance, instrumentation, conversation, popular music, vowel and consonant placement, breath management, social factors, and societies were brought together in an interactive multimedia platform, the "Global Jukebox" [13–18]. While several subsequent studies have used methods and samples modeled after Cantometrics to analyze hundreds of traditional music recordings from around the world [19–23], the full Global Jukebox sample of over 5,000 coded performances was never made publicly available, until now.

The Global Jukebox is an interactive online resource for exploring music and other performing arts cross-culturally. On it, the immediacy of field recordings representing the full range of the world's music can be experienced with reference to ethnographic, historical, environmental, linguistic, and geographic contexts, with song lyrics and testimonials by first-hand observers, musicians, and culture members. The Jukebox thus bridges the sciences and the humanities. It encompasses thousands of examples of singing, dancing, speaking, instrumentation, and other performing arts from over 1,000 societies, transformed into an online form that can be used for research, education, and cultural activism.

This article announces the long-anticipated publication of the raw coded data in downloadable form of Cantometrics and six additional studies and two supporting datasets (see Table 1), on https://theglobaljukebox.org. By releasing these data to the public, we hope to enrich scientific data on the expressive arts, to support cultural diversity, and to facilitate the practice of cultural equity in homes, classrooms, and research organizations.

## Background

Scientific cross-cultural comparison of music had been conducted since the late 19th century when the invention of the phonograph made relatively objective comparison of sound possible for the first time, birthing the field of comparative musicology, which later became known as "ethnomusicology" [25–30]. But until Alan Lomax's research on expressive culture, such comparisons were generally limited to relatively small samples of recordings and emphasized those aspects of melody, pitch and rhythmic structure that are privileged in Western staff notation. In intellectual partnership with the anthropologist Conrad Arensberg, Lomax worked with multidisciplinary teams from the 1960s through the mid-1990s to collect and analyze

**Table 1. Global Jukebox datasets included in this public release.**

| Dataset | Description | Variables | Performances/ Cases | Societies |
|---|---|---|---|---|
| **1. Cantometrics** https://zenodo.org/record/4898406 | Social and musical organization of singers, synchrony, relationships within and between vocal and instrumental parts, form | 37 | 5,776 songs | 1,026 |
| **2. Minutage** https://zenodo.org/record/4898387 | Phrasing and breath patterns in song | 36 | 687 songs | 118 |
| **3. Phonotactics** https://zenodo.org/record/4898383 | Vowel and consonant frequency patterns in singing | 51 | 338 songs | 47 |
| **4. Ensemble Study** https://zenodo.org/record/4898378 | Bibliographic information on ensembles worldwide | 12 | 776 ensembles | 153 |
| **5. Instrument Study** https://zenodo.org/record/4898389 | Bibliographic information and classification of instruments across ensembles worldwide | 14 | 1,780 instruments | 152 |
| **6. Parlametrics** https://zenodo.org/record/4898385 | Conversational style | 52 | 188 conversations | 158 |
| **7. Urban Strain** https://zenodo.org/record/4898365 | North American popular music and dance styles | 18 | 378 popular songs | 178* |
| *Supporting Datasets* | | | | |
| **Social Factors** https://zenodo.org/record/4898380 | Cross-cultural sample coded for geography, population size, subsistence, political structure, gender roles, kinship & family structure, property, social stratification, sexuality, games, theology (adapted and expanded from early versions of the Ethnographic Atlas) | 38 | 1,310 societies | 1,310 |
| **Societies (Cultures)** | Ethnographic descriptors for sampled societies. | 60 | 1,275 societies | 1,275 |

*To be categorized separately, since the classification of urban musical cultures requires a different scheme from the one used in cross-cultural research on traditional societies.

These datasets are stored in a series of CSV tables, within a cross-linguistic data format (CLDF) framework [24], which are released via Zenodo.

thousands of examples of recorded expressive traditions from all world regions with a new, radical approach that emphasized performance style and social interaction [14–18] (see 1.3 and S1.2 in S1 File for details).

Lomax and Arensberg treated the performing arts empirically, as expressive behavior, and searched for the recurring features of performance that affiliate and differentiate preferred styles of singing, dancing, and speaking in a culture or region. They theorized that culture is a web of interaction that is formalized and codified in the expressive arts. They proposed that performance styles are influenced by human fundamentals like subsistence, the organization of work, social structure, environment, and cultural history. They amassed large global samples of recorded performances and developed systems for coding them (Table 1). Each example is classified and coded by aesthetic, organizational, qualifying, and formal features that can be compared cross-culturally, making it possible to explore relationships between music, dance, speech, social life and the environment. A cross cultural approach modeled after the *Ethnographic Atlas* [1] was used to frame, sample, and analyze the resulting data; multivariate factor analysis and correlations tests were used to seek relationships and patterns. For example, they found large regions sharing similar performance styles, which were argued to match patterns of ancient human settlement, subsistence, and migration (Fig 1; see summary of more results in S1.2 in S1 File).

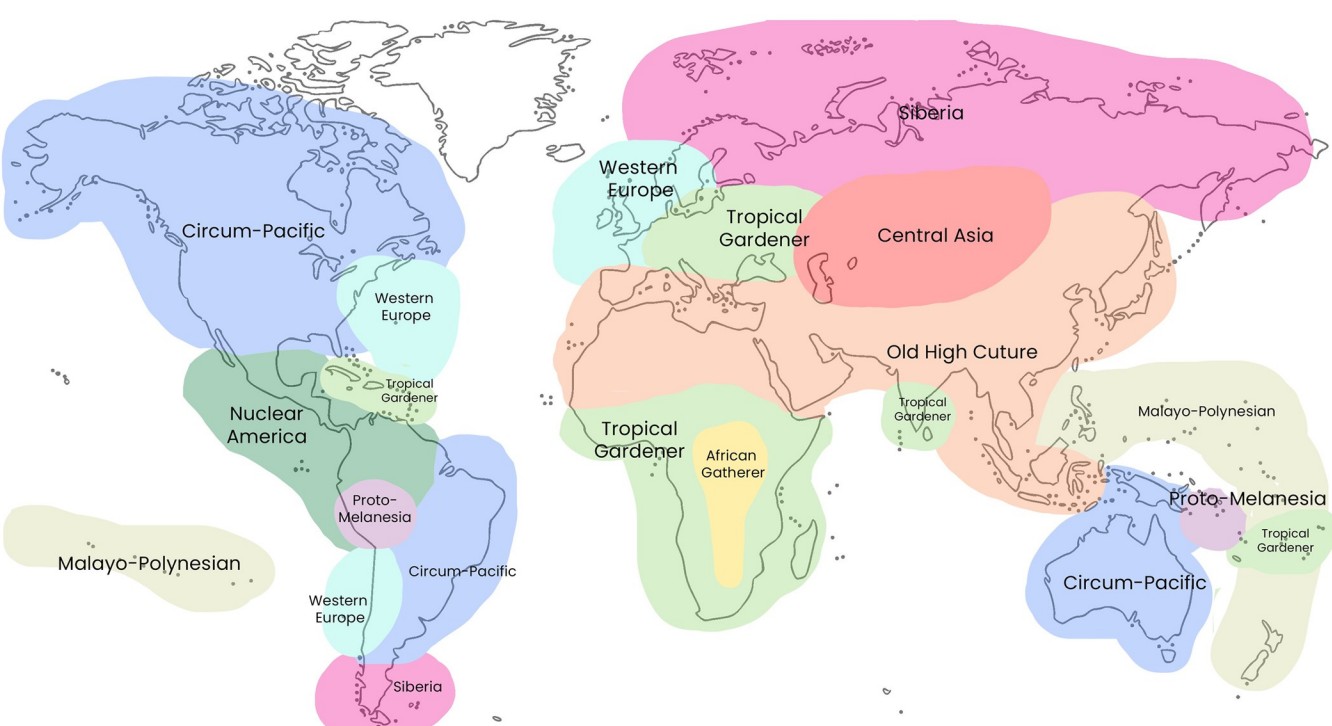

**Fig 1. Global map of 10 song-style regions previously identified by factor analyses of Cantometrics data (adapted from [15]).** Lomax and Arensberg's novel methods, execution, and conclusions drew considerable criticism when they were published and subsequently (e.g., [31–36]; cf. [14,17,18] for review and discussion). One major impediment to overcoming criticism was that the raw Cantometric data and sample were never made public for others to examine and reanalyze. The public release of the Global Jukebox now largely solves this problem.

## The Global Jukebox and its data

In the 1980s, the project's multiple datasets were incorporated into a single multimedia, relational database that Lomax called the Global Jukebox. Despite interest and financial support from Apple and other institutions, the project was curtailed by technological limitations and the intellectual climate of the eighties and nineties. In 2010, ALCW, an anthropologist, GD'A, a world leader in architectural digital design who had worked with Lomax on the Global Jukebox prototype, and a team of ethnomusicologists, designers and programmers, stepped in to realize the project.

A preliminary beta version of a new Global Jukebox, was released in 2017 as an online resource for exploring music and other performing arts cross-culturally. While the release generated much excitement among the public and media such as the *New York Times* and *Rolling Stone* [37–41], the digitized codings had not yet undergone quality control and were not yet downloadable for research or accompanied by any formal peer-reviewed documentation of the kind now provided in this article. Between 2017 and 2021, the raw data and metadata of Cantometrics were processed for full publication: descriptive data on each song and society were revised and expanded with song details, lyrics, genres and instruments, and nearly two thousand missing and truncated audio files were added. Songs were assigned to narrower and more accurate societal groupings with new coordinates. An updated comprehensive dataset of societies was constructed within a neutral geographic classification with linguistic, ethnohistorical, climatological and terrain referents. Societies were matched to the *Ethnographic Atlas* [1], the Standard Cross-Cultural Sample [42], the Binford Hunter-Gatherer dataset [43], and Jorgensen's Western North American Indian dataset [44] so that the database could be shared

with D-PLACE [5] (see S1.4.3 in S1 File for details). JM applied automated procedures for identifying potential coding errors to the Cantometrics data, which were then manually checked and corrected as necessary (see S1.8 in S1 File). Coding reliability was rigorously tested on a random sample (see S1.7 in S1 File). Finally, once we had completed and documented this multi-year quality-control process, we made the full Global Jukebox data and metadata downloadable via GitHub and Zenodo (https://github.com/theglobaljukebox) in May 2021 in tandem with uploading a preprint of the current manuscript to *PsyArXiv* (https://psyarxiv.com/4z97j).

Our article focuses primarily on the data of Cantometrics, and concludes with an important piece of reanalysis (see section 5). We announce the long-anticipated publication of the raw coded Cantometrics data in downloadable form, and of six additional studies plus two supporting datasets (see Table 1), representing over 1000 societies. This public release will make it possible for scientists to test cross-cultural hypotheses using global data, and will stimulate new research and debate on cross-cultural diversity, the nature of aesthetic preferences, and the role of expressive arts in human evolution [6–12,45]. It also supports the practice of cultural equity in scientific approaches to music research.

## 1. The data

### 1.1. Datasets of the Global Jukebox

Foundational to the Global Jukebox are its datasets. Seven cross-cultural studies on distinct but related aspects of the performing arts, and two supporting datasets of societies are currently available (see Table 1; full descriptions in S1.3.1 in S1 File). An additional five performance datasets, plus four derivative studies, and one supporting taxonomy of subsistence are in progress, and are not included in the current release (see S11 Table in S1 File). This article focuses on the largest, best known, and most comprehensive dataset, **Cantometrics**, which uses 37 features (cf. Table 2) to code musical style for 5,776 audio recordings of traditional songs from 1,026 societies.

We also introduce several datasets representing more detailed analyses of performance style using many of the audio recordings coded by Cantometrics, as well as additional recordings: *Minutage* uses 36 additional variables to code breathing and phrasing patterns for 687 songs from 118 societies, while *Phonotactics* uses 51 variables to code vowel and consonant use for 338 songs from 47 societies. In addition, we include the *Instruments* dataset, which uses 14 different variables to code structural and functional aspects of 1,780 instruments from 152 societies, *Ensembles* with 776 cases from 153 societies, the *Urban Strain* (a popular song study) with 378 songs and dances from North America, and the *Parlametrics* dataset, which uses 52 variables to code aspects of spoken conversation for 188 audio recordings from 158 societies (Table 1). Unlike Cantometrics, these datasets have not yet been thoroughly cleaned and checked, but are being released together for maximum accessibility and transparency. Other datasets in the process of being digitized and prepared for release in a future publication are *Choreometrics*, *Personnel and Orchestra*, *Song Texts*, *Vocal Qualities*, *Popular Songs* (an extension of the Urban Strain set not yet coded), and several derivative studies on song style and social structure (see S1.3.3 in S1 File for details).

Finally, we publish data on social structure ("Social Factors" and "Societies") built from an adapted version of the Ethnographic Atlas by Lomax, Arensberg, Barbara Ayres and their colleagues, and used in Lomax et al.'s previous analyses of relationships between music and culture. Their inclusion here will allow replication and extension of Lomax et al.'s original analyses.

**Table 2. Cantometric variables\*.**

| Line | Variable | Description |
|---|---|---|
| Line 1 | The social organization of the vocal group | Type of group organization employed by the vocal group, considering the prominence of solo or leader's part, measured on a scale of increasingly integrated social relations, from solo to interlock. |
| Line 2 | Relationship of orchestra to vocal parts | Size of the instrumental group and its relationship to the vocal part (accompanying, predominant, unrelated, or complementary). |
| Line 3 | Social organization of the orchestra | Type of group organization employed by the instrumental group, considering the prominence of solo or leader's part, measured on a scale of increasingly integrated social relations, from solo to interlock. |
| Line 4 | Musical organization of the vocal part | Type of harmonic and rhythmic coordination amongst singers. |
| Line 5 | Tonal blend of the vocal group | Degree to which singers match one anothers' vocal timbre. |
| Line 6 | Rhythmic coordination of the vocal group | Degree to which singers synchronize the attacks and releases of each note. |
| Line 7 | Musical organization of the orchestra | Type of harmonic and rhythmic coordination amongst instrumentalists. |
| Line 8 | Tonal blend of the orchestra | Degree to which instrumentalists match one anothers' timbre. |
| Line 9 | Rhythmic coordination of the orchestra | Degree to which instrumentalists in the group synchronize the attacks and releases of each note. |
| Line 10 | Repetition of text | Proportion of repeated words or non-lexical utterances in relation to unrepeated words in the song. |
| Line 11 | Overall rhythm: vocal | Type of meter used in the vocal part. |
| Line 12 | Rhythmic relationship within the vocal group | Rhythmic relationship between singers within the overall metrical pattern. |
| Line 13 | Overall rhythm: orchestra | Type of meter used in the instrumental part. |
| Line 14 | Rhythmic relationship within the orchestra | Rhythmic relationship between instruments within the overall metrical pattern. |
| Line 15 | Melodic shape | Most typical pitch contour of each phrase of the song. |
| Line 16 | Melodic form | Form of the entire song, considering the degree that melodic material is repeated (litany, strophe, or through-composed), the complexity of the form, and the degree of variation in each repeated section. |
| Line 17 | Phrase length | Length of a typical phrase in the song. |
| Line 18 | Number of phrases | Number of melodic phrases that occur in a song before a full repeat of the entire phrase sequence, and the presence or absence of phrase symmetry. |
| Line 19 | Position of final tone | Relation of the final sung note to the total range of the song's vocal part. |
| Line 20 | Melodic range | Interval between the highest and lowest sung notes of the song. |
| Line 21 | Interval size | Predominant interval between notes in the song's vocal melody. |
| Line 22 | Polyphonic type | Manner in which vocalists produce simultaneous intervals other than unison or the octave, if polyphony is present. |
| Line 23 | Embellishment | Degree to which singers embellish the basic melody of the song with more rapid and ephemeral notes. |
| Line 24 | Tempo | Speed of a performance. |
| Line 25 | Volume | Loudness of a performance. |
| Line 26 | Rubato: vocal | How strictly tempo is maintained during a vocal performance. |

(*Continued*)

**Table 2.** (Continued)

| Line | Variable | Description |
|------|----------|-------------|
| Line 27 | Rubato: orchestra | How strictly tempo is maintained by instrumentalists during a performance. |
| Line 28 | Glissando | Degree to which the singers' voices slides smoothly from one tone to another, passing through all the intermediate pitch levels. |
| Line 29 | Melisma | Degree to which the same syllable of text is sung to two or more notes of melody. |
| Line 30 | Tremolo | Degree of quavering or shaking in the singers' voices, heard as an undulation between two closely adjacent pitches or tone colors. |
| Line 31 | Glottal | Degree to which the vocal style includes forceful activity in the pharyngeal, or glottal, area in the back of the throat. |
| Line 32 | Vocal pitch (register) | Register or pitch range that the singers' voices fall into. |
| Line 33 | Vocal width | Measurement of tension in the vocal timbre, compares voices that are mellow, relaxed, and resonant (wide), with voices that are tense, pinched, and restricted in resonance (narrow). |
| Line 34 | Nasality | Degree of nasalization in the vocal part (when the soft palate drops and air is forced through the nose, producing a "twanging"). |
| Line 35 | Rasp | Degree of noise or stridency in the vocal timbre. |
| Line 36 | Accent | Degree of forcefulness applied to the dynamic attack on sung tones. |
| Line 37 | Enunciation | Degree of precision with which consonants are articulated in sung texts. |

*For a more detailed explanation of the definitions of the Cantometric variables and their coding criteria, see [46,47] and the online training examples in the "Songs of Earth" section (http://theglobaljukebox.org/?songsofearth). See S9 Fig and S12 Table in S1 File for inter-rater reliability data for each variable.

## 1.2. Comparison with other cross cultural datasets

One of the major advantages of the Global Jukebox is its size and global scope compared to other published cross-cultural performing arts datasets. In particular, the Cantometrics dataset of 5,776 coded songs from 1,026 societies is more than an order of magnitude larger than the largest previously available global datasets that used Cantometrics or similar coding schemes to classify musical style, such as the 304 recordings from the *Garland Encyclopedia of World Music* analyzed in [19] or the 118 recordings from the *Natural History of Song Discography (NHS-D)* analyzed in [20]. In addition to raw recording numbers, Cantometrics is also more balanced in many ways than NHS-D or Garland: Cantometrics includes more songs per society (median: 4 for Cantometrics vs. 1 for NHS-D and Garland), has better representation of small-scale societies than Garland, and better representation of large-scale societies than NHS. Of course, each dataset has its own sampling criteria and analysis goals, each with its own strengths and weaknesses: for instance, like the NHS-D, Cantometrics' principal focus is vocal music, but it includes several variables for the analysis of orchestras, and its data are linked with a dataset classifying personnel and orchestra (to be published soon), and two others classifying instruments and ensembles. Garland includes both instrumental and vocal music. At the same time, the NHS-D only sampled examples of four song genres (dance songs, lullabies, love songs, and healing songs), while Cantometrics and Garland include numerous traditional genres that are identified by their emic names as well as broader genre categories. Unlike Cantometrics and Garland, NHS-D includes transcriptions in Western staff notation in addition to Cantometric-like codings. Clearly, Cantometrics may be more suitable for some purposes

and not others, while all of these datasets may serve as complementary sources of evidence to investigate questions about cross-cultural musical diversity. For a fuller description of key differences between these three global datasets and other regional datasets, cf. [48]. (See also section 2.1 for more details on the Cantometrics sampling methodology).

## 1.3. Coded performance variables: Selection and reliability

Cantometric variables were developed through field observation, intensive listening, and experimentation. Lomax and his collaborators looked for performance qualities with significant worldwide variation and with a role in shaping performance traditions. Variables requiring fine distinctions to be made were discarded. Lomax and Victor Grauer [46] aimed to transcend Western staff notation's overly narrow focus on pitch and rhythm, and aimed to include variables capturing aspects of cohesiveness and differentiation within performances, social and musical organization, synchrony, relationships within and between instruments and voices, text, ornamentation, and vocal qualities. Together with melodic, rhythmic and structural features, these factors aimed to capture a broad matrix of aesthetic and social codes or conventions that are fairly consistent at the cultural level.

Following a similar methodology, specialist teams developed parameters for coding dance, speech and additional aspects of music, producing multiple datasets (see S3-S9 Tables in S1 File for variable descriptions for other published datasets). The inclusion of speech in this project, alongside forms more typically recognized as art, like song and dance, reflects Lomax's intentions to study the cross-cultural aesthetic patterns of expressive culture in its various forms rather than on the basis of conventional definitions of "art." Full details for most coding systems have been republished or published for the first time in [46,47]. The Global Jukebox website contains a fully digitized training course ("Songs of Earth: Aesthetic and Social Codes in Music"), with recordings and coding guides to allow researchers to interpret existing Cantometrics codings and to become trained Cantometric coders capable of adding new codings. New introductions, coding guides, and results for the other datasets are available in [47].

Many limitations of Cantometrics, Choreometrics, and related schemes have been previously critiqued (cf. [14,17,18] for review and discussion). For example, some argue they are too subjective, and thus have low reliability [32], while others argue they are not subjective enough in that they do not account for the ways in which similar sound structures can have different meanings in different cultures [33]. While we cannot provide data to evaluate all criticisms here, we did conduct an analysis of inter-rater reliability by having two trained coders (ALCW and PES) independently code a random sample of 30 songs from the Global Jukebox using Cantometrics and compare their codings against each others', against codings of students in Japan recently trained in Cantometrics, and against the codings in the Global Jukebox. These analyses found inter-rater reliability to be at acceptable levels on average, but with substantial variation in reliability across variables (see sections 4, S1.7, S9 Fig, and S12 Table in S1 File for details).

## 1.4. Performance data sources

Performance data is based on analysis of audiovisual sources—recorded examples of song and speech and filmed examples of dance and movement. As a documentarian himself, Alan Lomax had great faith in the recorded (and filmed) medium and what it could communicate. He already had a substantial library of world music on records and field tapes recorded himself or sent to him by colleagues, but he and Grauer spent a year acquiring more recordings to fill in the gaps of their first sample of over 2,000 recordings. Geopolitical tensions during the

middle of the Cold War limited the accessibility of some regions, and it was only possible later, in stages, to obtain material from some parts of Eastern Europe, the former Soviet Union, rural China, India, and the Pacific.

## 1.5. Selection of audio examples

Lomax primarily sampled folk and Indigenous songs for Cantometrics, although small samples of traditional art music and jazz are also included (e.g., Hindustani/Carnatic traditions of South Asia; Central European classical singing; various examples of Chinese opera, Javanese Gamelan, modern jazz, etc.). In his own research practice, Lomax sought out the oldest and most typical songs and performance styles because he believed they would shed light on how singing voices the touchstones of emotion and personality development in a culture [18]. He held extensive consultations with the singers and culture holders he recorded, which in many instances amounted to mini- or full autobiographical accounts (e.g., "Mr. Jelly Roll", a full-length curated autobiography [49] and Interviews with Bessie Jones [50]; see [51] for a book-length review). Representativeness within a tradition, performance quality, and aesthetic criteria were important in selecting songs for Cantometrics. Lomax made every effort to include women's and children's songs, although these weren't always readily available at the time. Final selections were based on subjective but informed decisions by Lomax and experts who recorded the songs. Lomax also drew upon his own extensive fieldwork, and from oral histories, recording liner notes, and the accounts of ethnographers and collectors.

An assumption of Cantometrics, based on Lomax's field experience, early experiments, and listening sessions with Victor Grauer (co-inventor of Cantometrics with Lomax), was that the same features of song style would appear throughout most examples of singing in a society. Extensive piloting during the development of the Cantometric song sample led Lomax and Grauer to conclude that approximately ten songs were sufficient to capture the key elements of a song style in most folk and Indigenous societies. In cases where it was not possible to code ten songs for a given society, Lomax's above-mentioned emphasis on representativeness of a tradition in the selection of song(s) to code justifies the inclusion of these society's data in cross-cultural comparisons and hypothesis testing. This assumption has been critiqued and it could be systematically retested, but it is debatable whether a better option was available given the limitations of available recordings and resources to devote to manually listening to and coding these recordings (cf. discussion in [14]). The broader criticism of insufficient song samples for some societies and regions, such as Polynesia, is valid and will be addressed in future updates to the data.

A related critique of the original Cantometrics dataset asserted that its song sample did not sufficiently cover intra-societal diversity, and that societal designations were too broad. These issues have been addressed in this current release by splitting some of the original societies into more specific subgroups, and assigning more precise ethnolinguistic and geographic designations to societies (see 2; S1.4 in S1 File). A consequence of this decision to classify societies at finer grain is that the database now appears to have a larger number of societies and smaller number of songs per society (currently 1,026 societies with a median of 4 songs per society, rather than the "minimum of ten songs per society" from 233 societies originally described by Lomax [14,16]. However, the current finer-grained classification allows researchers to model higher-order relationships between societies (e.g., linguistic, geographic) while also preserving potential differences between closely related societies. A recent study by Daikoku et al. [52] shows the potential to use Cantometrics data to investigate questions of intra-societal musical diversity.

### 1.6. Availability of audio recordings

Alan Lomax had research and publishing agreements with artists, collectors, filmmakers, repositories, and publishers; the Association for Cultural Equity (ACE) has obtained or requested streaming rights on the Global Jukebox website. But distributing these recordings presents challenges in terms of copyright and respecting the wishes of culture bearers. Our lawyers advise us to ask all remaining rights holders for permission to allow their songs to be downloaded for use in scientific research and publications. This is a lengthy and expensive process, and there is no way of circumnavigating it. The large institutions that now house many collections of field recording are increasingly reluctant to grant this kind of access to their holdings. We have excluded 456 Indigenous American and Australian songs from streaming pending permission from their communities. The audio files for 191 songs are currently missing, but their coded data is available.

However, we can make approximately 2000 of the songs available for download for scientific research and publication, and we have developed a process for facilitating the use of audio samples for research. We have prepared two agreements, which will be available on our website: (1) between researchers and the Global Jukebox with terms for using audio; (2) between third party rights holders to authorize researchers to use their recordings. APIs of audio examples will be linked with the song data on Github and Zenodo. To access them, an authorization token from the Global Jukebox will be required. Initially, approval will be handled case by case, but it can be automated in future if there is a sufficient volume of requests. Audio will be restricted to those samples ACE controls, with more added as rights holders opt in, or agree to the new terms through our ongoing efforts. Researchers who would like to use examples in scientific publication are asked to contact the rights holders or repositories for permission to use those songs; this will not be difficult with those samples ACE controls. Researchers can also link readers to a streaming audio playlist of selections on the Global Jukebox.

### 1.7. Performance metadata

Metadata for each coded song performance includes a unique coding identification number; source society; audio reference numbers and information; song and performer information; setting and context if known; collector, publishing and archival data and year recorded; comments; source tags; and additional descriptive information. Fig 2 is a screenshot of metadata and codings for one example song. Metadata for Cantometrics, Phonotactics, Minutage, Parlametrics, and Instruments and Ensembles are summarized in S10 Table in S1 File.

## 2. Sampling societies and links to other datasets

### 2.1. Sampling societies

The Jukebox is designed to be an experience of the expressive arts as *cultural* phenomena. The datasets (songs, instruments, conversation, etc.) are related through their source societies. We use the term "societies" as an alternative to the term "cultures" or other terms for defining cultural groups. The Global Jukebox follows the *Ethnographic Atlas* in adopting ethno-linguistically defined societies as a key unit of analysis. The Jukebox differs from the *Ethnographic Atlas* in that in most cases a given society is not represented by a single set of coded data, but instead contains multiple examples of different performances. Performances (songs, dances, etc.) thus represent a finer unit of analysis beneath the larger unit of societies, and make it possible to account for local/regional historical, sociological and aesthetic influences, as well as for musical diversity within societies [52,53]. Societies can also be related to one another at higher levels of organization (e.g., people, Koppen climate/terrain [54], language family, geographic

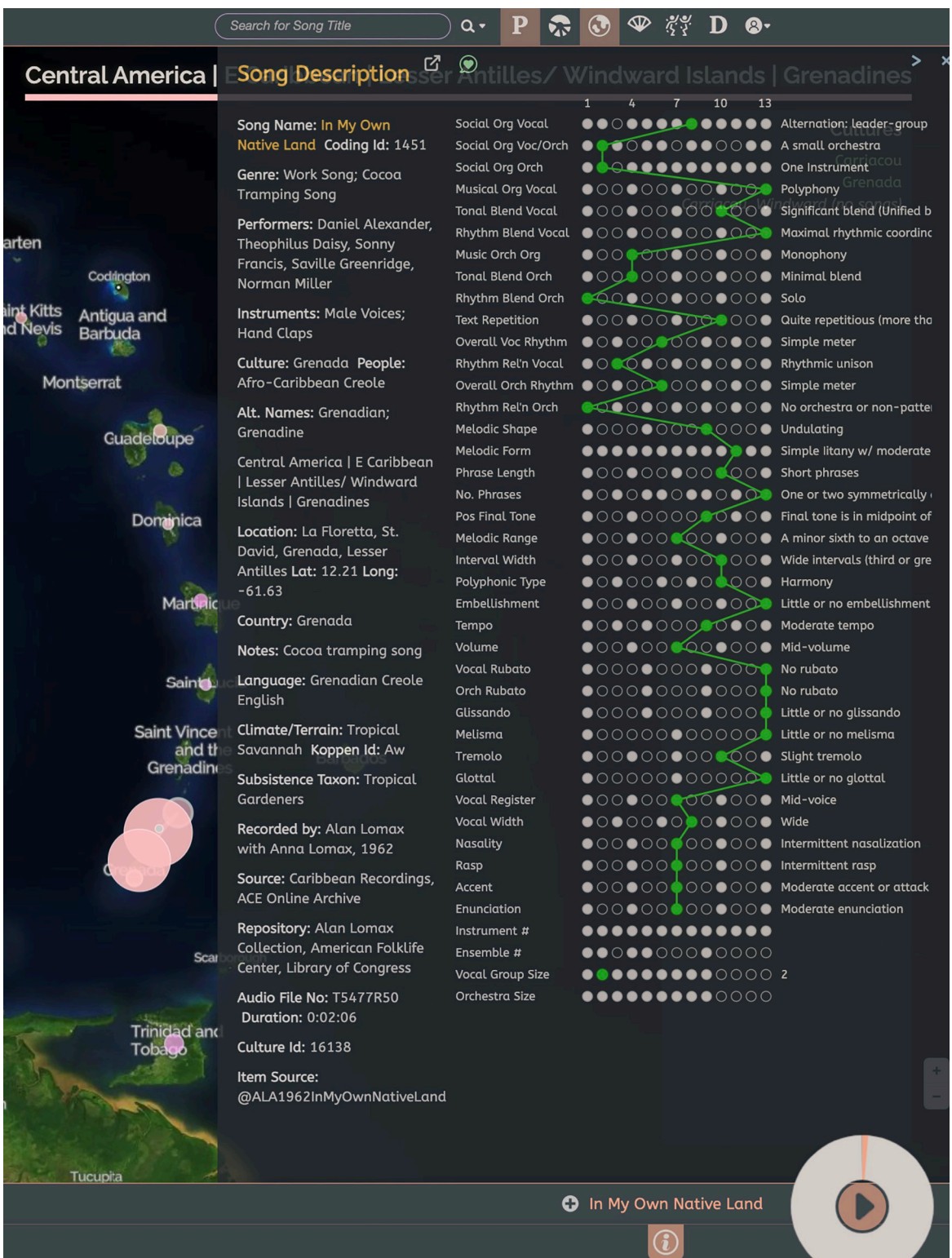

**Fig 2. Screenshot of metadata and Cantometric codings of the Grenadian song "In My Own Native Land" recorded in 1962 by Alan Lomax and ALCW.**

area or region, etc.; see S1.4.2 in S1 File). For visualization on the maps and linguistic trees of D-PLACE, songs and other performance data are condensed into a single "modal profile" representing the most common traits across all examples for a given society (cf. [5] for examples of how D-PLACE can be used to explore patterns in culture and their relationship to population history (via language) and geography).

A total of 1,275 Indigenous and folk societies across thirteen world regions are represented in the Global Jukebox, plus 472 popular song cultures. This accounts for societies that were sampled in the primary studies listed in Table 1, with the exception of Instruments and Ensembles. Where possible, Cantometrics was sampled from the more than 1,200 societies for which cultural data had already been coded in Murdock's *Ethnographic Atlas* to facilitate comparison of song style and social structure [14,17,18]. Of the total number of Global Jukebox societies, 1,234 are linked with coded data (including Choreometrics data coding dance style, which will be published in the future), and of that number, 1,026 are included in the Cantometrics set. In addition, 508 Cantometrics societies have been coded for a series of social variables, describing aspects of social and community organization, most of which are taken from (or slightly modified from) Murdock's Ethnographic Atlas variables. This dataset is released here as the 'Social Factors' dataset.

The cases included in the "Ensembles" and "Instruments" datasets depart from the other datasets in that they only partially conform to our definition of society. Because these studies use bibliographic sources rather than specific audio recordings, the societal designations made by the original investigators were often necessarily much broader than those in the other datasets (they 'lump' many societies that are 'split' in other datasets). Future research with scholars who specialize in musical instruments will be necessary to match the Instruments/Ensembles data with our more detailed societal data.

## 2.2. Links to other cross-cultural datasets

One goal of this release of Lomax's datasets is to enable researchers to map and reanalyze the kinds of relationships between the arts and society originally explored by Lomax (cf. Fig 4 for an example comparing distributions of song style and social complexity). Of the 1,026 societies sampled in the Cantometrics study, approximately half have been coded for other cultural features in one or more of the ethnographic datasets currently available through D-PLACE. Specifically, 469 Cantometrics societies are also coded in the Ethnographic Atlas; 66 in Binford's Forager Dataset; 122 in the Standard Cross-Cultural Sample; and 33 in Jorgensen's Western North American Indian Dataset. Over the past several years, efforts have been made to identify and provide links to the primary sources used to code cultural data in these historical cross-cultural datasets (see D-PLACE [d-place.org] [5]), making it easier for researchers to return to primary sources when a coded cultural variable of interest is not available. eHRAF World Cultures [ehrafworldcultures.yale.edu] [2] greatly facilitates this process, by providing finely subject-indexed, fully digitized and searchable primary ethnographic documents. Currently, 268 Cantometrics societies are covered by eHRAF.

## 3. Data curation, cleaning, and validation

Each of the 7 Global Jukebox datasets and the Social Factors supporting dataset listed in Table 1 have their own coding schemes and criteria. Fully cleaning each dataset takes a great deal of effort and resources, and we currently cannot fully clean all datasets for publication. We decided to focus our cleaning efforts on the largest and most influential dataset of 5,776 Cantometric codings of songs. We decided to simultaneously publish the other 6 partially cleaned datasets and the Social Factors supporting dataset listed in Table 1 in order to make

the materials available as widely as possible, but we emphasize that only the Cantometric data-set has been fully cleaned and validated. **S1.6-S1.8** in S1 File provide a detailed description of the data curation, digitization, cleaning, and validation process.

## 4. Coding reliability

Overall, our analyses suggest that both coding reliability (mean $\kappa$ = 0.54; S8 Fig and S12 Table in S1 File) and accuracy (approximately 0.4–1% rate of unambiguous coding/data entry errors; S9 Fig in S1 File) are at acceptable levels on average. However, there was also substantial variation in reliability across variables. Some variables showed near-perfect consensus: for example Line 4 differentiating solo vs. different types of group singing ($\kappa$ = 0.94 / 89% agreement) and Line 7 which captures similar information but for instrumental accompaniment ($\kappa$ = 0.92 / 86% agreement). Other variables had very low reliability effectively at chance levels (e.g., nasality [Line 33], vocal width [Line 32]). In general, aspects of vocal timbre (e.g., nasality, vocal width, rasp) tended to have lower reliability, but even within categories different variables could show strikingly different reliability. For example, Embellishment (Line 23) and Glissando (Line 29) are both intended to capture aspects of melodic ornamentation, but the former has much higher reliability ($\kappa$ = 0.70 vs. 0.19, respectively).

We caution that lower reliability does not necessarily mean a variable is "worse". For example, some variables with higher reliability may reflect more obvious acoustic features that could potentially be automated (e.g., presence vs. absence of instruments), while some variables with lower reliability could nevertheless reflect more subtle but meaningful features that require human annotation (e.g., melodic form). Finally, these ratings only reflect the codings of a few raters whose musical backgrounds are not representative of the world. We encourage users to use the reliability data as simply one of many pieces of evidence to guide decisions about variables for use in cross-cultural analysis, development of automated signal processing algorithms, or other uses.

## 5. Is song style correlated with social complexity? An example of hypothesis testing using the Global Jukebox

### 5.1. Hypothesis testing with the Global Jukebox datasets

Lomax and colleagues published numerous analyses of Cantometrics and other Global Jukebox data. These results were reviewed by Wood [17,18] and Savage [14], and are summarized in S1.2 in S1 File. Unfortunately, the original analyses cannot be precisely reproduced, as it is not possible to reconstruct the specific datasets and procedures that were used for previous analyses performed during the mid-20th century before data/code-sharing capabilities were widely available. For reanalyses of Cantometric data and analyses of similar data, see [19,21,22,52,53,55–61].

Lomax' original analyses consisted of bivariate correlations between all 37 musical variables and the dozens of social variables coded in the Ethnographic Atlas (EA). Lomax summarized five of these key proposed correlations between song style variables and social variables as follows:

*Song style tends to grow more articulated, ornamented, heavily orchestrated, and exclusive as societies grow bigger, more productive, more urbanized, and more stratified. Specifically, (1) the level of text repetition decreases directly as productivity increases, (2) the level of precision of enunciation increases as states grow in size, (3) the prominence of small intervals and embellishments indicates the level of stratification, (4) orchestral complexity symbolizes state*

*power, and (5) melodic form and complexity reflect the size and subsistence base of a community.* [62]

Understandably, readers may be skeptical about any attempt to establish predictive or causal relationships between society and music. Whether such correlations reflect causal relationships, autocorrelation due to shared ancestry, or something else, has been greatly debated [14–18,31–36,61]. Here we attempt to see whether the basic correlations are a) reproducible, and b) artifacts of geographic or linguistic autocorrelation.

## 5.2. Methods

A given society may contain stylistically similar or diverse sets of songs with similar or differing codings. Such codings can be condensed into a single "modal profile " in order to analyze relationships at the level of a society, as was the method in the original analysis, or clustered and analyzed separately as individual songs [21–23,55–60,63]. An example modal profile is mapped in Fig 3 for the Cantometric variable "Embellishment". To maximize usability, we have provided the data in a form treating songs as the unit of analysis (see cldf/songs.cv in https://zenodo.org/record/4898406)) and using modal profiles to treat societies as the unit of analysis (output/converted_modal_profiles.csv at https://zenodo.org/record/6537663#. YnszmllS_BK)).

Below, we use the society-level modal profiles to test 5 bivariate relationships between musical style and patterns of social structure, using cultural data from the Ethnographic Atlas [1] and linguistic and spatial data from D-PLACE [5]. These statistical tests revisit 5 key hypotheses proposed and tested by Alan Lomax using modal profiles, and reanalyze one of Lomax's

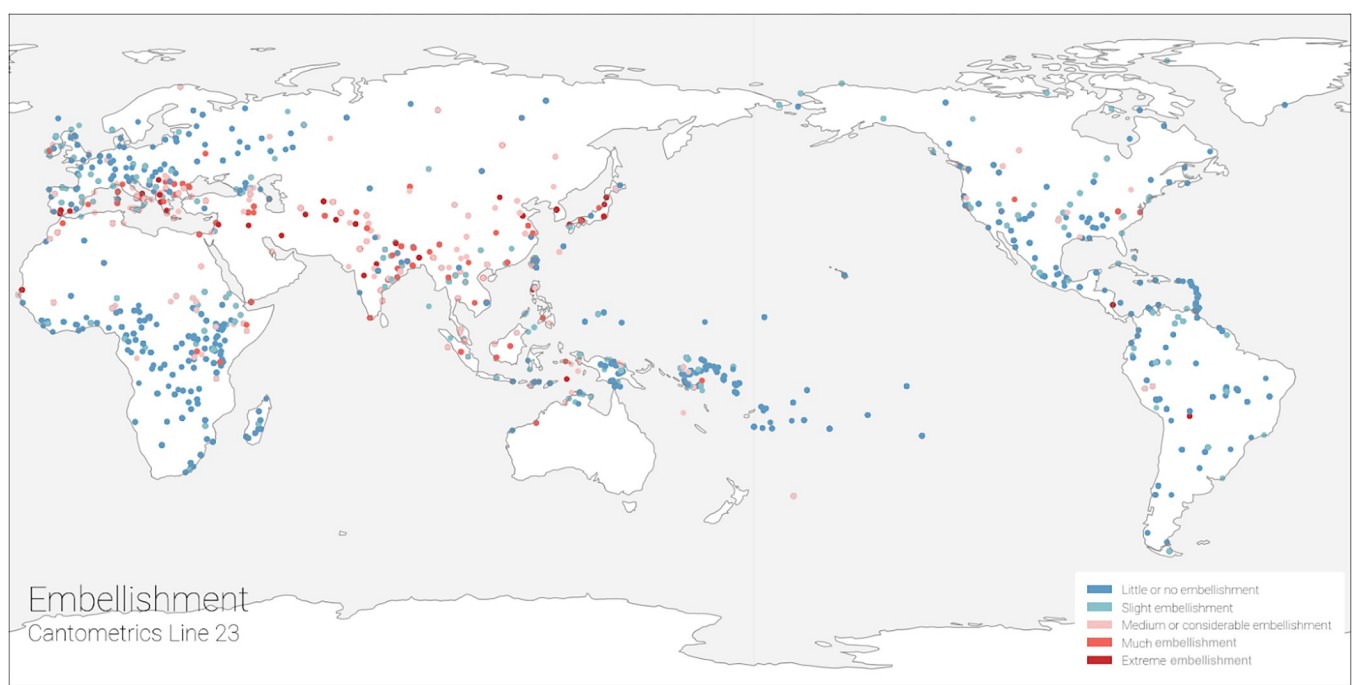

**Fig 3. An example map showing the global distribution of modal codings for one of the 37 Cantometric variables: CV23 ("Embellishment").** Embellishment is a technique in which rapid, ephemeral notes ornament the main melodic line, but are distinct from it. The distribution of highly embellished singing (in red tones) outlines the"Silk Road" region of Eurasian cultural exchange which includes the Mediterranean, North Africa, the Arabian Peninsula, the Middle East, Western Asia, South Asia, Southeast Asia, and East Asia. This highly embellished singing is differentiated from Eastern and Central Europe, sub-Saharan Africa, Oceania, and the Americas where singing is mostly unembellished (blue tones).

primary conclusions of his original Cantometric analyses: that global song style is correlated with social complexity [15,16].

A common critique of early Cantometrics analyses was that it did not control for common cultural patterns of autocorrelation [14,31]. Specifically, that the statistical evidence for the correlations occurred because societies were historically related, or frequently interacted with each other, rather than because of a functional relationship between music and social structure [64,65]. Here, we re-test the proposed correlations, as well as the aggregate complexity relationship, while controlling for historical relationships (using language), and spatial relationships (using geography).

We use three types of models in this analysis: first, a simple linear model. The simple linear model replicates methods used in the original Lomax tests and acts and a baseline model. Second, vertical relationships are tested using a phylogenetic linear regression, implemented in phylolm [66]. Finally, spatial relationships are tested within generalized mixed-models with non-gaussian random effects and exponential spatial correlation, using spaMM [67]. Phylogenetic relationships are determined from the glottolog taxonomy [4] with a Grafen branch length transformation, as performed in [68]. All Cantometrics and Ethnographic Atlas data were also standardized; see details in S1.9.1 in S1 File. For detailed definitions of the social and musical variables, see Table 2; S9 and S18-S28 Tables in S1 File.

The five correlations are between:

1. Musical organization of the Orchestra (CV7) ~ Jurisdictional hierarchy beyond the local community (EA033)

2. Text repetition (CV10) ~ Subsistence (an aggregate variable described in the SM);

3. Embellishment (CV23) ~ Class (EA066) + Caste (EA068) + Slavery (EA070)

4. Melodic interval size (CV21) ~ Community size (EA031), and

5. Enunciation (CV37) ~ Jurisdictional hierarchy beyond the local community (EA033)

Within each hypothesis, AIC model comparison allows us to determine whether the distribution of data is best explained by linguistic or geographic relatedness (or neither). Secondly, we test the general hypothesis of a relationship between average song style and social complexity by performing two principal component analyses (PCA) on the set of musical variables and the set of social variables used in the preceding tests. Using the first principal component from each variable set, we test the relationship between musical style and social complexity under the same three conditions as the bivariate models. More details of the analyses are available in section S1.9 in S1 File of the supplementary material.

## 5.3 Results of reanalysis

Across the five models, we find that all hypothesized correlations hold in at least one of the three model variations. Specifically: musical organization increases with more jurisdictional hierarchy; text repetition declines with more productive subsistence technologies; embellishment increases with stratification; melodic intervals decline with larger community size; and enunciation becomes more precise with higher levels of jurisdictional hierarchy. However, we also find that models that control for geographic relatedness improve model fit by greater than 2 AIC units in the Musical Organization of the Orchestra, Text Repetition, Embellishment, and Melodic interval Size models. We cannot statistically determine whether vertical or horizontal processes best explain variation in Enunciation (S17 Table in S1 File). These results suggest that the way we make music is guided by the musical traditions of our ancestors and our neighbors, and is also related to societal structure.

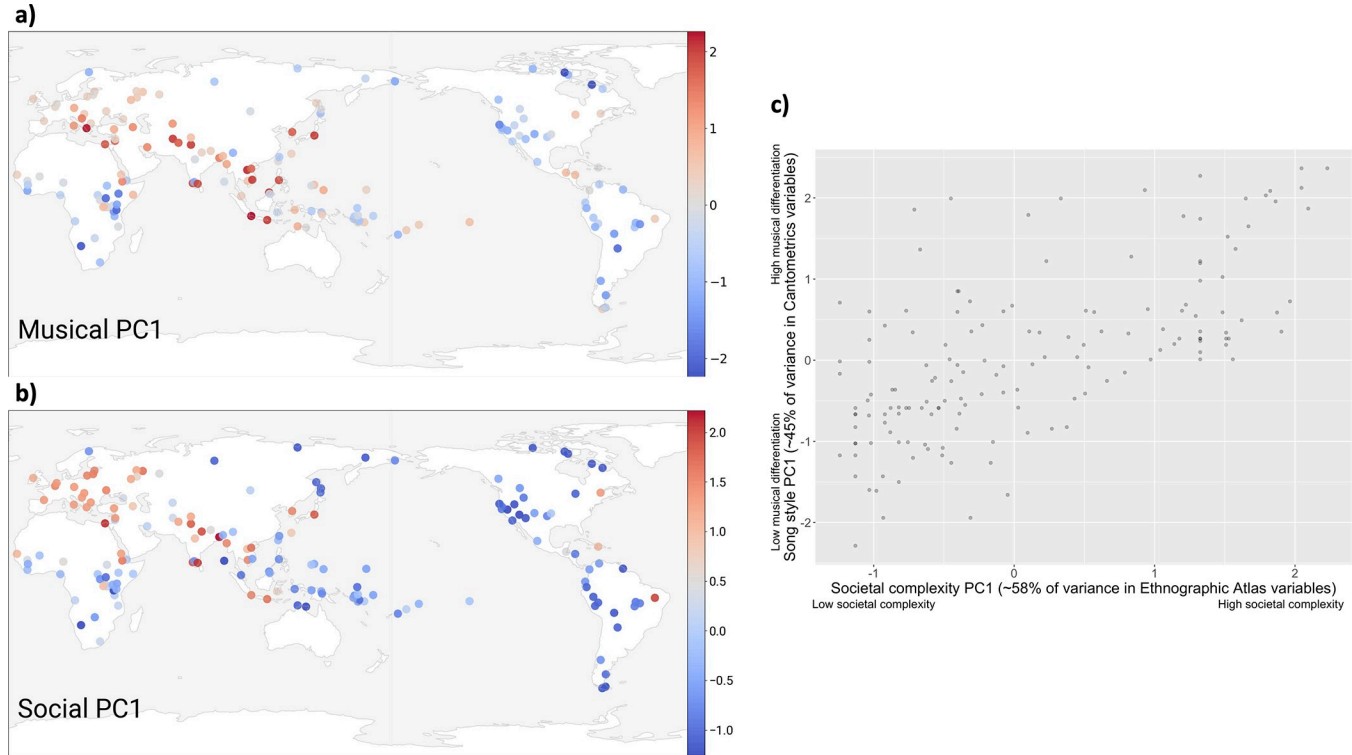

**Fig 4. A comparison of n = 135 societies with fully matched data for five Cantometric variables of musical style and seven Ethnographic Atlas variables of social complexity.** a) A map of the global distribution of the first principal component (PC1; interpreted as "musical differentiation") for five musical variables from the Global Jukebox's Cantometrics dataset (Musical organization of the Orchestra (CV7), Text repetition (CV10), Embellishment (CV23), Melodic interval size, and Enunciation (CV37). b) a) A map of the global distribution of the PC1 (interpreted as "societal complexity") for six social variables from D-PLACE's Ethnographic Atlas dataset (Jurisdictional hierarchy beyond the local community (EA033), Subsistence (an aggregate variable described in the SM); Class (EA066), Caste (EA068), Slavery (EA070), and Community size (EA031). c) The correlation between musical PC1 and social PC1 was significant when controlling for possible autocorrelation using linguistic relatedness and geographic proximity (see Supplementary Material S1.9 in S1 File for modeling details and analyses of bivariate correlations between the musical and social variables).

Using the eigenvalue criterion and visual inspection of scree-plots, we determined that both the musical and social variable sets are each best explained by a single dimension. The first principal component of musical variables explains 45% of variation, and the first principal component of social variables explains 58% of variation (proportions comparable to those found in previous analyses using similar [though not identical] variable sets [9,52]; for loadings see S15 Table in S1 File). We find a significant positive relationship between the two principal components, regardless of linguistic or geographic controls (Fig 4). However, AIC comparison revealed that a model controlling for geography best explained the data ($\beta = 0.60$, $p < 0.001$, $n = 147$; S13 Fig, S17 Table in S1 File).

### 5.4 Discussion

Lomax's correlations between song style and social complexity have been disputed [14–18,31–36,61], but our bivariate and PCA reanalyses provide evidence suggesting that his original correlations are a) reproducible, and b) not clear artifacts of geographic or linguistic autocorrelation. We do not believe, nor is there any evidence that Lomax believed, that social factors directly produce effects upon music. Here we do not attempt to apply formal causal analysis or test any proposed causal mechanisms. (cf. [61]). However, the probability that certain social and musical traits and configurations will consistently cooccur raises questions about what

drives aesthetic preferences. For example, are aesthetic preferences embodied in vocal representations of emotions and / or physical states that may arise under certain persistent conditions?

Different evolutionary theories propose several possible mechanistic pathways that could be tested [61], such as patterns of interaction [69], roles of social bonding [70], signaling [71], and psychosocial effects [16]. Our database can generate new hypotheses with preliminary evidence for causal and other kinds of relationships; these could then be tested by modeling archival information and data from the field. It can take ethnomusicological work beyond the descriptive by identifying and explaining such causal relationships, and it can introduce ethnologically grounded and cross-cultural data, methods and perspectives to cognitive and neuro-scientific research on music. We hope that future work with these open datasets will engender fresh insights into cultural evolution and the role of art and human expressivity in society.

## 6. Ethics, rights and consent

The Global Jukebox was created to provide a platform for the world's music and for the data to speak for itself, so that users can appreciate a diverse range of performative approaches, not by Western standards, but by listening to global samples and identifying their many approaches to performance [46,72]. This approach to cross-cultural performance analysis was in part based upon Lomax's extensive fieldwork with singers and musicians, where he recognized distinctive aesthetic and social values in their performances. Guiding his efforts was the conviction that every society's music must be appreciated and researched on its own terms, according to its own aesthetic standards, given equitable airplay, and play a significant part in education [cf. [73–75].

The Association for Cultural Equity (ACE) is committed to obtaining permission to stream the media examples that have been studied and analyzed for the Global Jukebox. As did Lomax, ACE seeks out the estates of artists recorded by Lomax, and their descendents and estates receive fees and royalties from licensing and sales. Repatriation of Lomax's recordings to their communities of origin, in partnership with those communities, is ongoing and has reached over 50 communities, descendents of artists, and national libraries. North American and Australian Indigenous audio samples will be streamed on the Jukebox only with the agreement of each community.

To improve ethical practices, ACE convenes with cultural advocates from diverse communities [76,77]. ACE's online resources are among the very few, if not the only, ones that stream media examples in their entirety (except those on Spotify, in which case users must sign up for Spotify to hear more than 30 seconds), and have no sign in or membership requirements for using them. To further improve access to Lomax's recordings and research, ACE engages with community arts leaders, artists and other culture bearers to connect their constituencies to the Global Jukebox and our online archive in meaningful ways. They are invited to contribute Journeys and Exhibits, correct metadata, interpret the songs, suggest new songs and codings, and add their documentation to the songs. We plan to work with culture-bearers to expand and improve the Global Jukebox sample and data. ACE also supports emerging leaders from endangered cultures to document, describe, steward and reappraise their expressive traditions. Additional information regarding the ethical, cultural, and scientific considerations specific to inclusivity in global research is included in the Supporting Information (S1 Checklist).

## 7. Conclusion

The full publication of Global Jukebox data represents the culmination of sixty years of research by one of the world's most influential scholars of music [51,75]. We are making these

data publicly available in order to encourage their use, improvement, and expansion through diverse intercultural and interdisciplinary collaborations. We also hope to encourage further scientific research into music, dance, speech and other arts as primary rather than ancillary factors in human history and evolution, as well as to deepen our understanding of cross-cultural diversity at a time when it is more important than ever before.

## How to cite the Global Jukebox

Research that uses data from the Global Jukebox should cite both the original source(s) of the data and this paper (e.g., research using data from the Cantometrics dataset: "Lomax (1968, 1980); Wood et al. (2022)"). The reference list should include the date that data was accessed and URL for the Global Jukebox (http://theglobaljukebox.org), in addition to the full reference for Lomax (1968). Additionally, each dataset (Table 1) is versioned and stored on Zenodo. Users can cite the specific dataset and version used by visiting https://zenodo.org/record/4898406.

## Supporting information

**S1 Checklist. PLOS's questionnaire on inclusivity in global research.**
(PDF)

**S1 File. Supplementary material.** This document includes all additional information and supplementary tables and figures referenced in the manuscript. This includes material regarding original data analysis and results, details of the Global Jukebox datasets, information on societies, data history, data normalization, coding reliability, data validation, current analyses, and the concept of cultural equity.
(PDF)

## Acknowledgments

We thank the thousands of musicians, ethnomusicologists, record labels, and funding agencies whose decades of work resulted in the audio recordings that provide the foundation of the Global Jukebox (see Lomax 1968:xv-xvii for a full list of Acknowledgments, and see meta-data at http://theglobaljukebox.org for detailed credits for each song; see Fig 2 for an example). Alan Lomax, with Michael del Rio, imagined and realized the essence of the Global Jukebox. It was based on Lomax's years of fieldwork and experimentation, and thirty years of research on performance style with Conrad M. Arensberg and specialist collaborators. George P. Murdock, whose foundational work in cross-cultural anthropology led to the Ethnographic Atlas, and Norman Berkowitz, a cutting edge programmer and statistician, each played a leading role in operationalizing theories and secondary hypotheses. Richard Smith brought the website out of the dustbin of obsolete programming languages and obsolete hard drives. Gideon D'Arcangelo helped to bring the Jukebox to life as it is now being developed. Other contributors include Jeff Feddersen (Original Design); Ray Cha (Wireframe); Alona Weiss (Additional Design); Kiki Smith-Archiapatti (Design and Content); Martin Szinger (Developer); Steve Rosenthal and the Magic Shop (Audio Digital Transfers, Restoration); Forrestine Paulay (Choreometrics); Karen Kohn Bradley, Frederick Curry, Meriam Lobel, Sinclair O'Gaga, Onye Ozuzu, Miriam Philips, Susan Wiesner (Dance Analysis and Social Justice); Patricia Campbell (Curriculum); Todd Harvey, Jorge Arevalo Mateus, Bruno Nettl, Anthony Seeger, Michael Tenzer, Philip Yampolsky (Ethnomusicology Advisors), Karen Claman, Kathleen Rivera (Researchers); Miriam Elhajli (Latin American folk sample); Jesse Rifkin, Don Fleming (Popular Song); Victor Grauer (Cantometrics Advisor); Sergio Bonanzinga, Judith Cohen, Lamont Pearly III, Mark Slobin (Journeys); Herb Sturz.

The Global Jukebox is a project of the Association for Cultural Equity (culturalequity.org), a 501c(3) non-profit charitable organization and custodian of the Alan Lomax Archive. Founded by Alan Lomax in 1983, ACE's mission is to stimulate cultural equity through fostering research, dissemination, and sustainability of the world's traditional expressive practices. It endeavors to reconnect people and communities with their creative heritage through open access and mutual engagement. Lomax's original recordings and papers were deposited with the American Folklife Center of The Library of Congress; ACE retains digital copies which it uses in repatriation, publication, and collaborative initiatives with source communities.

## Hunter College of CUNY

ACE is grateful to Hunter College of CUNY and Jennifer Raab for their generous support and for giving a home to the Global Jukebox and the Lomax archive.

## Author Contributions

**Conceptualization:** Anna L. C. Wood, Kathryn R. Kirby, Carol R. Ember, Sam Passmore, John McBride, Gideon D'Arcangelo, Patrick E. Savage.

**Data curation:** Anna L. C. Wood, Kathryn R. Kirby, Stella Silbert, Sam Passmore, Hideo Daikoku, John McBride, Forrestine Paulay, Michael J. Flory, John Szinger, Gideon D'Arcangelo, Karen Kohn Bradley, Marco Guarino, Maisa Atayeva, Jesse Rifkin, Violet Baron, Miriam El Hajli, Martin Szinger, Patrick E. Savage.

**Formal analysis:** Sam Passmore, Hideo Daikoku, John McBride, Michael J. Flory.

**Funding acquisition:** Anna L. C. Wood, Patrick E. Savage.

**Investigation:** Anna L. C. Wood, Stella Silbert, Sam Passmore, Hideo Daikoku, John McBride, Jesse Rifkin, Patrick E. Savage.

**Methodology:** Anna L. C. Wood, Stella Silbert, Sam Passmore, Hideo Daikoku, John McBride, Patrick E. Savage.

**Project administration:** Anna L. C. Wood, Patrick E. Savage.

**Software:** Sam Passmore, Hideo Daikoku, John McBride, John Szinger.

**Supervision:** Anna L. C. Wood, Patrick E. Savage.

**Validation:** Stella Silbert, Sam Passmore, John McBride, Patrick E. Savage.

**Visualization:** Stella Silbert, Sam Passmore, Hideo Daikoku, John McBride, Patrick E. Savage.

**Writing – original draft:** Anna L. C. Wood, Stella Silbert, Sam Passmore, John McBride, Patrick E. Savage.

**Writing – review & editing:** Anna L. C. Wood, Kathryn R. Kirby, Carol R. Ember, Stella Silbert, Sam Passmore, Hideo Daikoku, John McBride, Forrestine Paulay, Michael J. Flory, John Szinger, Gideon D'Arcangelo, Karen Kohn Bradley, Marco Guarino, Maisa Atayeva, Jesse Rifkin, Violet Baron, Miriam El Hajli, Martin Szinger.

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
