## [Decision Letter · Decision Letter 0]

19 Sep 2022

The Global Jukebox: A Public Database of Performing Arts and Culture

PONE-D-22-15328

Dear Dr. Savage,

I have now received reports from two independent reviewers who are experts in their respective fields. I'd like to thank the reviewers for their thoughtful comments and for taking on the assignment.

As you will read, the reviewers were very supportive of the manuscript in its current form, and both felt it warranted publication in PLOS One. Similarly, I find the work to be substantive, original, and of high quality. Therefore, I am pleased to inform you that your manuscript has been judged scientifically suitable for publication and will be formally accepted for publication once it meets all outstanding technical requirements.

Reviewer 1 had a minor suggestion which I'd ask you to please consider incorporating when making your final proof edits. As your manuscript has been under review here and elsewhere for an extended period of time, I felt it appropriate to accept at this stage and leave this edit to your discretion.

Kind regards,

Steven R Livingstone

Academic Editor

PLOS ONE

Journal Requirements:

1. Please include a complete copy of PLOS’ questionnaire on inclusivity in global research in your revised manuscript. Our policy for research in this area aims to improve transparency in the reporting of research performed outside of researchers’ own country or community. The policy applies to researchers who have travelled to a different country to conduct research, research with Indigenous populations or their lands, and research on cultural artefacts. The questionnaire can also be requested at the journal’s discretion for any other submissions, even if these conditions are not met.  Please find more information on the policy and a link to download a blank copy of the questionnaire here: https://journals.plos.org/plosone/s/best-practices-in-research-reporting. Please upload a completed version of your questionnaire as Supporting Information when you resubmit your manuscript.

3. We noted in your submission details that a portion of your manuscript may have been presented or published elsewhere. Please clarify whether this publication was peer-reviewed and formally published. If this work was previously peer-reviewed and published, in the cover letter please provide the reason that this work does not constitute dual publication and should be included in the current manuscript.

4. We note that you have referenced (ie Bewick et al. [5]”) which has currently not yet been accepted for publication. Please respond by return e-mail with a copy of your updated manuscript to include to remove this from your References and amend this to state in the body of your manuscript: (ie “Bewick et al. [Unpublished]”) as detailed online in our guide for authors

http://journals.plos.org/plosone/s/submission-guidelines#loc-reference-style.   We can then upload this to your submission on your behalf.

Reviewers' comments:

Reviewer's Responses to Questions

**Comments to the Author**

1. Is the manuscript technically sound, and do the data support the conclusions?

Reviewer #1: Yes

Reviewer #2: Yes

2. Has the statistical analysis been performed appropriately and rigorously? 

Reviewer #1: Yes

Reviewer #2: Yes

3. Have the authors made all data underlying the findings in their manuscript fully available?

Reviewer #1: Yes

Reviewer #2: Yes

4. Is the manuscript presented in an intelligible fashion and written in standard English?

Reviewer #1: Yes

Reviewer #2: Yes

5. Review Comments to the Author

Reviewer #1: This is a peculiar manuscript, and I was positively surprised for the amount of work it contains. I’m fascinated with the history described by the Authors and I believe (very personal opinion) this is a kind of work which can be accepted as it is or disliked. Since I’m fascinated, I am happy to endorse its publication, with just one remark.

As a psychologist with a peculiar interest in music perception, I would like to suggest only a possible specific adding: in the Discussion, the Authors rise the interesting question about “what drives aesthetic preferences” in humans, in a cross-cultural view. This is a milestone in the psychology of music field, and in this regard, I want to highlight some possible references which can add important views in this work. For instance, a study suggested that the right half of the brain would be specialized in processing music consonance, and this result can be intended as an evolutionary predisposition to appreciate certain types of sounds worldwide (https://pubmed.ncbi.nlm.nih.gov/25256169/; see also https://pubmed.ncbi.nlm.nih.gov/29547746/), together with some psychological relaxation effects induced by certain “classical” (Western) pieces (e.g., https://pubmed.ncbi.nlm.nih.gov/35753309/). However, a cross-cultural study revealed that music preferences are attributable to music exposure, thus to cultural rules – somewhat excluding a general and innate preference (https://pubmed.ncbi.nlm.nih.gov/32704441/). I think this diatribe could be a useful hint in a similar work.

Typo: line 575. Is there a typo before the reference n. 69?

Reviewer #2: Very valuable data that should have a large impact in the ethnomusicology research community.

The Global Jukebox makes available much of the Cantometrics data. Despite being already quite an old data it is still very relevant.

6. PLOS authors have the option to publish the peer review history of their article (what does this mean?). If published, this will include your full peer review and any attached files.

Reviewer #1: No

Reviewer #2: No

---

## [Editor Report · Acceptance letter]

11 Oct 2022

PONE-D-22-15328 

The Global Jukebox: A Public Database of Performing Arts and Culture 

Dear Dr. Savage:

I'm pleased to inform you that your manuscript has been deemed suitable for publication in PLOS ONE. Congratulations! Your manuscript is now with our production department. 

Kind regards, 

on behalf of

Dr. Steven R Livingstone 

Academic Editor

PLOS ONE